# Characterization of mangrove mudflat sediment contamination by fecal bacteria and trace metals: A multivariate assessment in equatorial marine protected areas in Gabon, Western Central Africa

Johann Ludovic Martial Happi[1,2], Aimé Roger Nzigou[1], Gauthier Schaal[3], Marie-Laure Rouget[4], Rolf Gael Mabicka Obame[1], François Le Loc'h[3], Jean-Daniel Mbega[5], Christophe Leboulanger[2]*, Patrick Mickala[1]

1 Université des Sciences et Techniques de Masuku, Franceville, Gabon, 2 MARBEC, Université de Montpellier, CNRS, Ifremer, IRD, Sète, France, 3 Univ Brest, CNRS, IRD, Ifremer, LEMAR, Plouzané, France, 4 Institut Universitaire Européen de la Mer, IUEM, UAR, Univ Brest, CNRS, IRD, Plouzané, France, 5 Laboratoire d'Hydrobiologie et d'Ichtyologie, IRAF, CENAREST, Libreville, Gabon

* christophe.leboulanger@ird.fr

## Abstract

Among the most productive ecosystems worldwide, mangroves contribute to global carbon sequestration and play a pivotal role for many species, including supporting the sustainable provision of intertropical fisheries resources. Despite the many essential ecosystem services they provide, mangrove ecosystems are facing increasing anthropogenic pressure, primarily because they develop in littoral areas where human activities are rapidly expanding, causing deforestation, urban extension and pollution. Mangroves cover over 1,700 km² of Gabon's coastline, stretching within protected zones or located alongside populated areas, raising concerns about the potential impact of pollution. This study assessed pollution levels in mangrove surface sediments from 24 stations in the capital city of Libreville and the two adjacent Marine Protected Areas (MPAs), Akanda and Pongara, where human impact has not been fully characterized. From 2021 to 2023, we measured chlorophyll *a*, fecal indicator bacteria (FIB: *Escherichia coli* and fecal streptococci) and six trace metals (As, Cd, Cr, Cu, Pb and Zn). Our results show that while Pongara National Park remains relatively pristine, human influence has resulted in significant contamination of sediments by FIB and trace metals, with Cu, Pb and Zn being associated with boat traffic and fuel spills. Chlorophyll *a* at the surface of sediments, indicative of microphytobenthic biomass, reflect the eutrophication from urban discharge. Spatial differences in contamination patterns are significant between Libreville and Akanda, and between Akanda and Pongara parks (p < 0.05), but not significant between Libreville and Pongara National Park (p > 0.05), suggesting a possible rapid dispersion of urban pollution. Despite localized pollution, contamination levels remained generally low, suggesting that patterns are influenced more by the specific characteristics of each

**Data availability statement:** All data used in the present sudy are fully available at the IRD DatVerse repository (French National Research Institute for Sustainable Development), https://doi.org/10.23708/GN6XDK.

**Funding:** Johann L.M. Happi benefited from a PhD stipend through the ARTS Program of IRD (French National Research Institute for Sustainable Development). The Funder had no role in study design, data collection and analysis, decision to publish, or preparation of the manuscript.

**Competing interests:** The authors have declared that no competing interests exist.

area than by MPA status. This study provides a first assessment of pollution pressure on mangrove tidal flats within protected areas in Gabon, emphasizing that the success of MPAs could be enhanced through contamination monitoring to achieve long-term conservation goals.

## Introduction

In the 21st century, protected areas have been promoted as a key strategy component of biodiversity conservation [1], yet a quarter of them are located within 20 km of urban areas questioning their effectiveness. Since 2002, Gabon has protected 11% of its land through a network of thirteen terrestrial national parks (NPs), followed by the designation in 2014 of 26% of its Exclusive Economic Zone (EEZs) as Marine Protected Areas (MPAs). This currently exceeds international commitments for MPA coverage compared to countries with a similar or higher economic status [2]. In Gabon, roughly 89% of the population lives in the two coastal urban areas of Libreville and Port-Gentil, a figure that is expected to rise up to 95% by 2050 [3].

The capital, Libreville, the most densely populated city in Gabon with 3,700 inhabitants per km$^2$, is bordered by three protected areas constitutive of the so-called the "Emerald Ark": Pongara National Park (PNP) to the south, Akanda National Park (ANP) to the north-east, and the coastal forest reserve of the Raponda Walker Arboretum (ARW) to the north [4]. Both national parks are internationally recognized for their importance to migratory shorebirds and comprise diverse coastal wetlands covering 745.9 km$^2$ in PNP and 464.5 km$^2$ in ANP. These wetlands include flooded forests and savannahs, herbaceous wetlands, mudflats, and mangrove forests [5]. Mangroves are dominant ecosystems in the two parks, harboring the tallest mangrove trees reported to date [6], with six species distributed across the Rhiziphoraceae, Avicenniaceae and Combretaceae families [7]. They also provide valuables services, mainly by sustaining resource for local artisanal fisheries [8,9], providing habitat for endangered large vertebrates (African manatee, leatherback, green and hawksbill sea turtles, Atlantic humpback dolphin), and contributing to carbon sequestration [10]. However, the proximity of the protected areas to the capital city exposes them to increasing anthropogenic pressures due to population growth and Gabon's extensive extractive industries, including offshore oil exploration and manganese and gold mining [11]. This occurred in the ARW coastal forest reserve between 1951 and 2004, submitted to degazetting [12] in spite of the presence of 24 endemic plant species [4] mostly due to land conversion to urban uses. This urban pressure is responsible for 56–96% of the measured deforestation in mangroves around Libreville, raising the annual rate of mangrove decay to 0.8% in the area, compared to a mean rate of 0.11% worldwide [13]. Other studies have documented additional environmental pressures, including microbiological water contamination [14], and persistent organic pollutants and trace metals occurrence in marine organisms [15]. However, knowledge about contamination levels of surface sediments in mangrove tidal flats in the Emerald Ark area is still unavailable.

Sediment contamination is typically caused by inorganic and organic that enter mangrove ecosystems mainly through industrial discharges, domestic wastewater, leaching from landfills and plantations, and mining activities [16]. These pollutants have the potential to cause irreversible adverse effects on ecosystems and are also a threat to human health. The importance of considering sediments for pollution assessment is driven by the continuous interactions between water, biota, and the complex but stable system they form [17]. On the other hand, depending on their properties, coastal sediments are recognized as reservoirs for various contaminants, including trace metals (TMs) [18,19] and fecal indicator bacteria (FIBs) [20]. Fecal indicator bacteria, such as *Escherichia coli* (*E. coli*) and fecal streptococci (Strept), originate from the gastrointestinal tract of humans and vertebrates. They tend to be more persistent in sediments than in the water column and are commonly used to assess microbiological water pollution [21].

Trace metals are inorganic elements released into the coastal environment from both natural sources and anthropogenic activities. On the one hand, several trace metals, like copper and zinc, are necessary for organism metabolism at trace concentrations but can result in toxicity if present in excessive levels. On the other hand, elements such as cadmium and lead have no known biological role and are toxic to organisms [22]. Biological and metallic contaminants are commonly released together with nutrients (mainly nitrogen and phosphorus) resulting in eutrophication of waters. Chlorophyll *a* (Chl *a*), a proxy of phytoplankton and benthic microalgae biomass, can be used as indicator of such pressures. In mudflats, Chl *a* surficial density in surface sediments reflects overall photosynthetic productivity of microalgal biofilms [23] that can significantly contribute to ecosystem mangrove carbon burial [24].

Because the different contaminants reflect different contamination pathways, assessing their concentrations using integrative approaches is a relevant way to understand the pressures on coastal sediments. Since the establishment of the protected areas around Libreville in 2002, no studies have yet been conducted on contaminants in their sediments. Moreover, no evaluation of the effectiveness of the MPA status in mitigating organic and inorganic pollutants in sediments has been carried out. The objective of the present study was to provide a comprehensive characterization of the current contamination levels in the mudflat sediments of two MPAs, compared with those in the urban area of Libreville. We specifically aimed at: (i) establishing a baseline for all stakeholders regarding chlorophyll *a*, fecal indicator bacteria, and trace metal concentrations in mangrove sediments, (ii) using sediment quality guidelines (SQGs) and indicators such as Enrichment Factor EF and Geo-accumulation Index $I_{geo}$, to assess both the potential biological impacts on marine organisms and contamination sources of metal pollution, and (iii) evaluating the extent and nature of sediment contamination in these MPAs. The overarching hypothesis of the study posits that MPAs are not fully effective in mitigating all pressures and that sediments within protected areas exhibit contamination levels comparable to those in areas near urban centers.

This information will also make a valuable contribution to the development of future conservation policies and waste management for coastal mangrove ecosystems and protected areas across Central Africa, especially considering the challenges posed by the rapid growth of urban populations.

## Materials and methods

### Study area

Mudflat sediment contamination levels were addressed within three main areas: Pongara National Park (PNP) on the south bank of the Komo estuary, Akanda National Park (ANP) in Mondah Bay, and the Libreville urban area located between the two estuaries (Fig 1, S1 Table in S1 File). Pongara NP covers an 890 km² area, of which approximately 55% consists in mangrove forest [25]. It is considered relatively pristine, as it is separated from the city of Libreville by the 10 km-wide Komo estuary. Akanda NP covers 540 km², including 46% of estuarine and sea water areas and 33% of mangrove forest [5], and is connected to Libreville via the Ntsini Channel. The Libreville area includes small mangrove stands bordering the city's channel network. These mangroves are likely under stronger anthropogenic influences, including agricultural, domestic, and industrial activities, than those in the parks. The whole region is under equatorial tropical climate

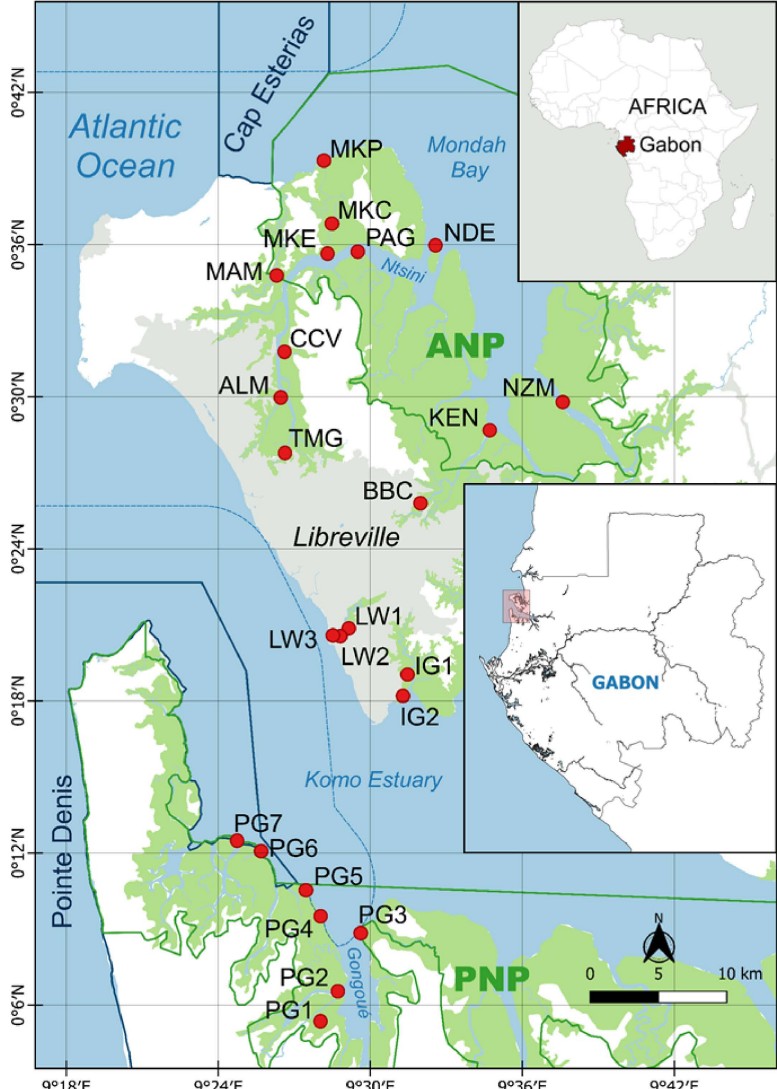

**Fig 1. Study area and sampling sites.** Akanda (ANP) and Pongara (PNP) National Parks are Filedelineated by solid green lines while Marine protected areas are outlined by solid blue lines. Buffer zones are represented by dashed lines and the urban area in grey. Sampling stations are indicated by red dots, three-digit code established from local names when available and station numbers (see S1 Table in S1 File for reference). Map from JLMH's personal work using Qgis 3.22 freeware, based on openstreetmap.org and marineregions.org, supplemented by publicly available data [27, 28, 29]. Contains information from OpenStreetMap and OpenStreetMap Foundation, which is made available under the Open Database License.

with high rainfall, reaching close 4000 mm annually north of Libreville, with a short dry season from June to August [26]. The mudflats of ANP cover 14.38 km$^2$ while those of PNP cover 10.29 km$^2$, and both are subject to tidal fluctuations [5].

## Sampling procedures

Four sampling campaigns were performed at low tides between September 2021 and July 2023 (S2 Table in S1 File) across estuarine gradients to capture contamination variability, both in urbanized zone and protected areas. A total of twenty-four sampling stations were selected: seven within PNP, eight within ANP, and nine within Libreville. At each station, the 5 mm top layer of sediment was collected for benthic Chl *a* quantification and FIB enumeration by pooling three

replicate samples of 46.75 cm$^2$ each, the 0–3 cm layer cleared of debris for trace metal analysis [30], and 0–20 cm cores preserved using a 3 cm diameter PVC corer for sediment characterization. In situ sediment temperature at 10 cm below surface was measured with a Hanna Checktemp HI98501 thermometer (Hanna Instruments). Immediately after collection, wet sediment samples were placed in an icebox for transportation to the laboratory. The samples intended for trace metal analysis were stored at −18°C until further processing, whereas the samples for biological analysis were analyzed promptly upon arrival.

### Sediment properties

At each station, two sediment samples were collected down to 20 cm. The grain size distribution was determined using the wet method [31], using stainless-steel sieves with mesh sizes of 63 µm, 125 µm, 250 µm, 500 µm, and 1 mm. Sediment total organic matter contents were evaluated using loss on ignition (LOI) at 450°C for 4 h in a muffle furnace [32] and reported as percentages (%). Sediment pore water salinity was measured on supernatant after centrifugation (5 minutes, 2000 rpm) with a Hanna HI99301 conductimeter (Hanna Instruments). Pore water dissolved nutrients ($NH_4^+$ and $PO_4^{3-}$) were measured spectrophotometrically using commercial kits (Spectroquant, Merck) as described in Leboulanger et al. [14] (S3 Table in S1 File).

### Chlorophyll *a* analysis

The density of Chl *a* in the surface sediment top layer, expressed as mg Chl *a*.m$^{-2}$, was measured using the methods described by Plante-Cuny [33]. An acrylic box was used to delineate three 5.5 x 8.5 surfaces, gently skimmed using a scalpel, and thoroughly mixed before processing. The total wet weight (corresponding to 3 x 46.75 = 140.25 cm$^2$) was measured to ensure further correspondence of subsamples with *in situ* sediment surface. Three aliquots of approximately 3 g of sediment were weighed and extracted using 9 mL of 96% ethanol in 13 ml polyethylene centrifuge tubes. Chlorophyll extraction was completed using ultrasonication for 30 seconds in an ice bath, followed by overnight at −18°C in the dark. Then, the samples were centrifuged and the supernatant absorbances measured throughout the visible light spectrum (380–800 nm) using a *SpectroDirect* spectrophotometer (AquaLytic). Following absorbance measurements and solvent removal, each sediment was dried at 60°C for 72 hours and weighed again to determine dry mass and water content. Chlorophyll *a* pigments density in the surface sediment were calculated based on formulas developed by Ritchie [34]: the concentrations of chlorophyll *a* were expressed in µg.L$^{-1}$ in the ethanol extract and then normalized to the sampling surface area, accounting for the mass of the extracted subsample, wet weight of total top layer sample, and total extracted volume (ethanol + sediment water content). Mean concentrations of the aliquots were calculated and the results for surface chlorophyll density reported in mg.m$^{-2}$. Blank controls of the concentration measurements were performed using ethanol (96%).

### Fecal indicator bacteria analysis

Two types of fecal indicator bacteria (FIB), fecal streptococci (Strept) and *Escherichia coli* (*E. coli*), were isolated from the sediment using the method previously described by Ghaderpour et al. [20]. Sediment aliquots of approximately 1 g for each targeted bacterium were weighed and homogenized in 9 ml of sterile 0.9% saline solution. After centrifugation two volumes of 4 ml of the supernatant were filtered through a sterile 47-mm polycarbonate squared membrane. Ready-to-use dehydrated selective medium kits AZIDE and ENDO (Sartorius) and specific incubation conditions [14], were used to quantify Strept and *E. coli*, respectively. The results are reported per 100 ml of filtered solution, with the average expressed in colony-forming units (CFU) per 100 ml volume of extract. When the filters were too heavily laden with merging colonies, an arbitrary number was assigned, considering the maximum number of colonies counted across all filters in the study as 10 000 CFU. Quality control of the analytical procedure was performed using 100 ml of commercial mineral water as blank.

## Trace metal analytical procedure

The concentration of trace metals (TMs) was measured using Inductively Coupled Plasma Mass Spectrometry (ICP-MS) with a Mixed Acid Digestion method. An amount of 100 mg of each powdered dry sediment sample was digested in aqua regia ($HNO_3$/HCl, 1:3 M) in pre-cleaned Savillex beakers, heated at 70°C for 1 hour, then at 90°C for 48 hours. A second digestion with $HNO_3$/HF (1:2 M) followed, with evaporation and a final dissolution in 10 mL of 6N HCl. The analyses were conducted on a Thermo Scientific ELEMENT XR™ spectrometer at the "Pôle Spectrométrie Océan," IUEM, Plouzané (France).

To prevent contamination, all equipment was acid-washed and dried at 90°C. Analytical accuracy was checked using geochemical certified reference materials (CRM JSD-2), with recovery rates averaging 95.36% (min 95%, max 103.2%) for As, Cd, Pb, Zn, and Fe, 87% for Cr and 89% for Cu.

## Trace metals contamination assessment

To assess the ecotoxicological hazard of TMs against benthic organisms, the Sediment Quality Guidelines (SQGs) established by NOAA [35] were used, including the TEL/PEL and ERL/ERM: the Threshold Effect Level (TEL) and the Effects range-low (ERL) represent the levels under which adverse effects of a contaminant are unlikely, while the Potential Effect Level (PEL) and the effect range-median (ERM) represent the concentration value beyond which a TM commonly produces negative effects to benthic organisms [35,36]. The difference between the two sets is that ERL/ERM represented the lower 10th and the 50th percentiles of the effects data [35] whereas TEL/PEL used the geometric mean of the aforementioned percentiles and incorporates effects plus no effects data [37,38].

To evidence anthropogenic TM contamination in superficial mudflat sediments, the Enrichment Factor (EF) and the Geo-accumulation Index ($I_{geo}$) were calculated. The EF [39] identifies anthropogenic contributions to metal pollution for each element [40] according to the formula:

$$EF = (C_i/C_m)_{sample} / (C_{bi}/C_{bm})_{background}$$

where $C_i$ and $C_m$ are the target TM and reference metal concentrations measured in the sample respectively, and $C_{bi}$ and $C_{bm}$ their respective background values. Manganese (Mn) and iron (Fe) are commonly used as reference metals [40], and Fe was selected in the present study according to its predominantly geological origin [41]. The EF is classified into 7 levels: EF < 1 indicates no enrichment, 1 < EF < 3: low enrichment, 3 < EF < 5: moderate enrichment, 5 < EF < 10: relatively rich enrichment, 10 < EF < 25: severe enrichment, 25 < EF < 50: very severe enrichment, and EF > 50: extremely rich enrichment.

The $I_{geo}$ index [42] quantifies metal accumulation relative to background levels according to:

$$I_{geo} = Log_2 C_{i\ sample}/1.5C_b)$$

where $C_i$ is the measured TM concentration, $C_b$ its background value, and 1.5 a correction factor for lithogenic inputs. Computed $I_{geo}$ values are interpreted as follows: < 0: unpolluted, 0–1: unpolluted to moderately polluted, 1–2: moderately polluted, 2–3: moderate to highly polluted, 3–4: highly polluted, 4–5: highly to very polluted, and > 5: very highly polluted.

$I_{geo}$ is not affected by geological factors and is suitable for the comparison of the same metal in different sampling sites while considering the influence of sediment texture, whereas EF is more suitable for the comparison of different metals in the same sampling sites [43]. As there are no existing pre-industrial background values for the study area, the average shale values from sedimentary rocks [44] were used as geochemical background values (Table 1). Six TMs were targeted according to their potential toxicity to living organisms: Cd, Pb, Cu, Cr, As, and Zn.

**Table 1. Trace metals geochemical background. $C_b$ values reported in µg.g$^{-1}$ for each of the six metals.**

| | As | Cd | Cr | Cu | Fe | Pb | Zn | References |
|---|---|---|---|---|---|---|---|---|
| Geochemical background values $C_b$ | 13 | 0.3 | 90 | 45 | 47,200 | 20 | 95 | [45] |

## Data analysis

Data from all campaigns have been used whenever available. R statistical software [45] and the *knn* function of VIM package [46] were used to impute missing data when data were not available. The *knn* function is used to aggregate the *k* value of the nearest neighbors as the imputed value. ANOVAs and Kruskal-Wallis tests were performed on the three sites to test for significant differences in contaminant concentrations, while Student's t-test and Wilcoxon tests were used to compare pairwise levels of contamination. Spatial distribution and relationships between biological markers, trace metals and sediment characteristics were examined using principal component analysis (PCA) and Pearson correlations. As the prerequisites for multivariate analysis of variance (MANOVA), such as the multivariate normality test, are difficult to verify, we used nonparametric multivariate analysis (npmv) provided by the npmv R package [47]. In contrast to classical MANOVA, npmv provides both the global significance between groups and the significance of subgroups of the response variables. In addition, npmv identifies the levels of the factors that are responsible for the global significance [48]. Prior to the Pearson correlation between the environmental factors, a logarithmic transformation was performed on all parameters to improve the normality of the data distribution [49]. The statistical analysis and the description of the grain size distribution were carried out with the R package G2Sd [50] and the ternary plot design with the R package ggtern [51]. Visualization of statistical analysis and graphics were designed with the ggplot2 package [52].

## Ethics statement

The research authorization and the work permit in protected areas were granted respectively by CENAREST review board (*Comité National des Autorisations de Recherche*, CNAR) and the Scientific Unit of ANPN (Gabonese National Agency for National Parks).

## Results

### Sediment properties

Sediment profiles of the three sites were determined based on sand, mud (clay + silt) and organic matter content (Fig 2). The percentage of sand is very high at LBV, ranging from 16 to 97%, while ANP has the lower percentage range (6.6 to 70.8%). The distribution of silt content was heterogeneous in LBV, but more homogeneous in PNP and ANP. There was no significant difference between LBV and PNP for sand (t-test: p = 0.071) while the difference was statistically significant between LBV-ANP (p < 0.001) and ANP-PNP (p = 0.005) (Fig 2A–2B). Organic matter (OM) content ranged from 0.2 to 29.9% in LBV, from 15.5 to 33.6% in ANP with four outliers (3.5, 3.8, 74 and 37.9%) and from 7.7 to 41.7% in PNP. ANOVA shows a statistically significant difference in %OM between the three sites (p = 0.017), particularly between LBV-ANP (p = 0.004) (Fig 2C).

The sediment profiles of LBV and PNP appear to be similar, whereas they differ from those of ANP. In both PNP and LBV, the sediment composition and texture ranges from muddy sand to sandy mud, with a dominance of fine and very fine sand in LBV. In contrast, ANP is dominated by silt and has a texture that ranges from muddy to sandy mud (Fig 3).

### Biological pollution markers

The distributions of Chl *a* and FIB among the three sites are shown in Fig 4 (S4 Table in S1 File). The chlorophyll *a* (Chl *a*) concentration ranged from 10.94 to 145.93 mg.m$^{-2}$ in LBV, from 13.2 to 84.81 mg.m$^{-2}$ in ANP and from 25.57 to 100.12

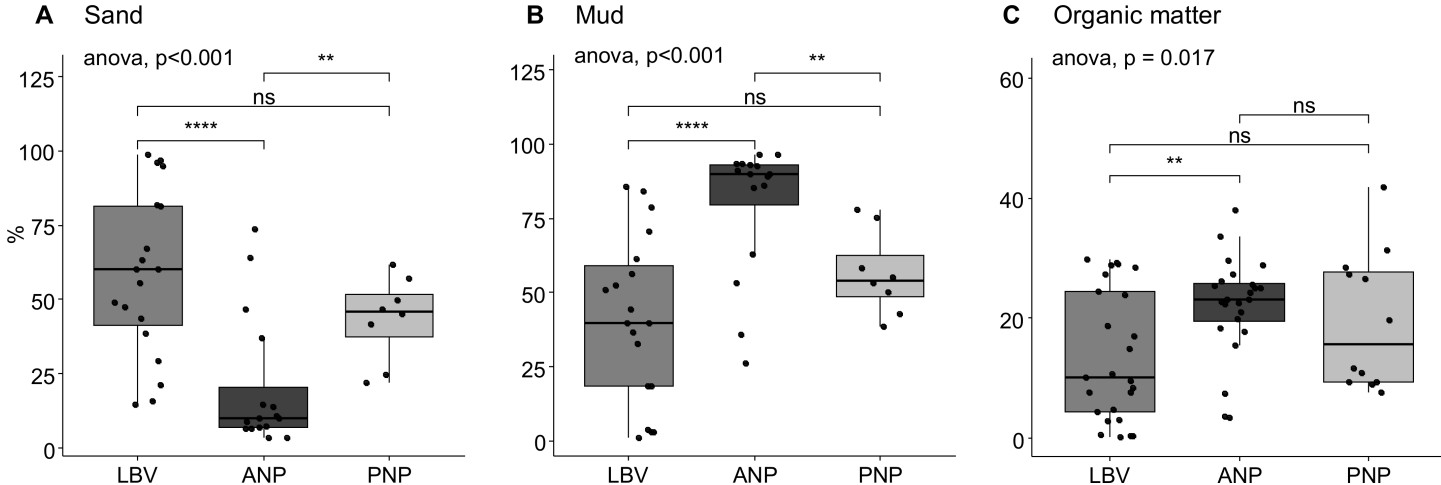

**Fig 2. Sediment composition.** Weight/ weight content in Sand (A), Mud (B) and Organic matter (C) are expressed as % of total. Samples are grouped according to their collection area, Libreville (LBV), Akanda NP (ANP), and Pongara NP (PNP). ANOVA p-value is given for the three groups differences; inter-group significant differences (t-student test) are marked with asterisks (*: $p < 0.05$; **: $p < 10^{-3}$; ***: $p < 10^{-4}$; ****: $p < 10^{-5}$). ns = non-significant difference.

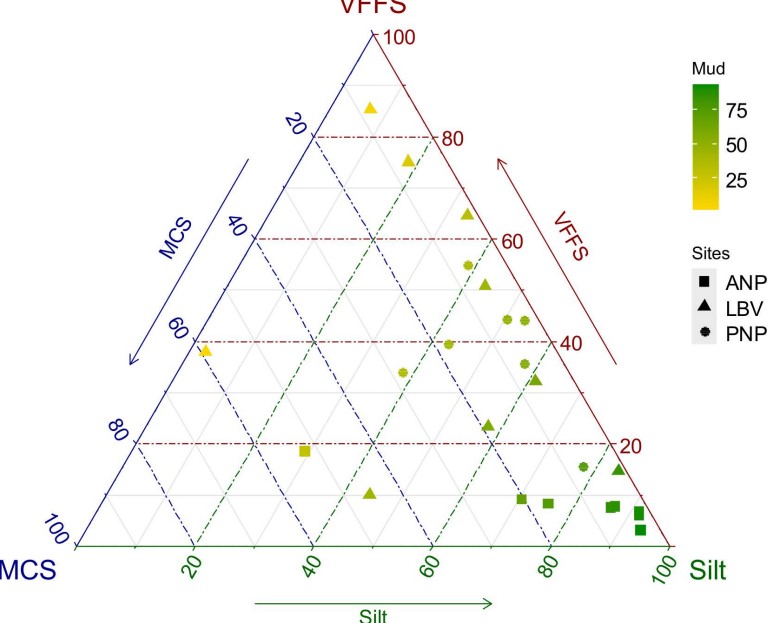

**Fig 3. Sediment texture.** Physical description of sediment texture, using ternary plot, in Akanda NP (squares), Pongara NP (circles), and Libreville (triangles) sampling sites. Color gradient from sand (yellow) to mud (green), and distribution according very fine and fine sand (VFFS), medium and coarse sand (MCS), and silt.

mg.m$^{-2}$ in PNP (Table 2). ANP and PNP showed lower and less variable Chl *a* surface densities, while Libreville showed the highest values and the greatest variability (Fig 4A). In Libreville, the highest Chl *a* surface densities were measured at urban stations near the harbor (92.45±52.66 and 72.09±26.74 mg.m$^{-2}$) and at the outlet of the canal that discharges

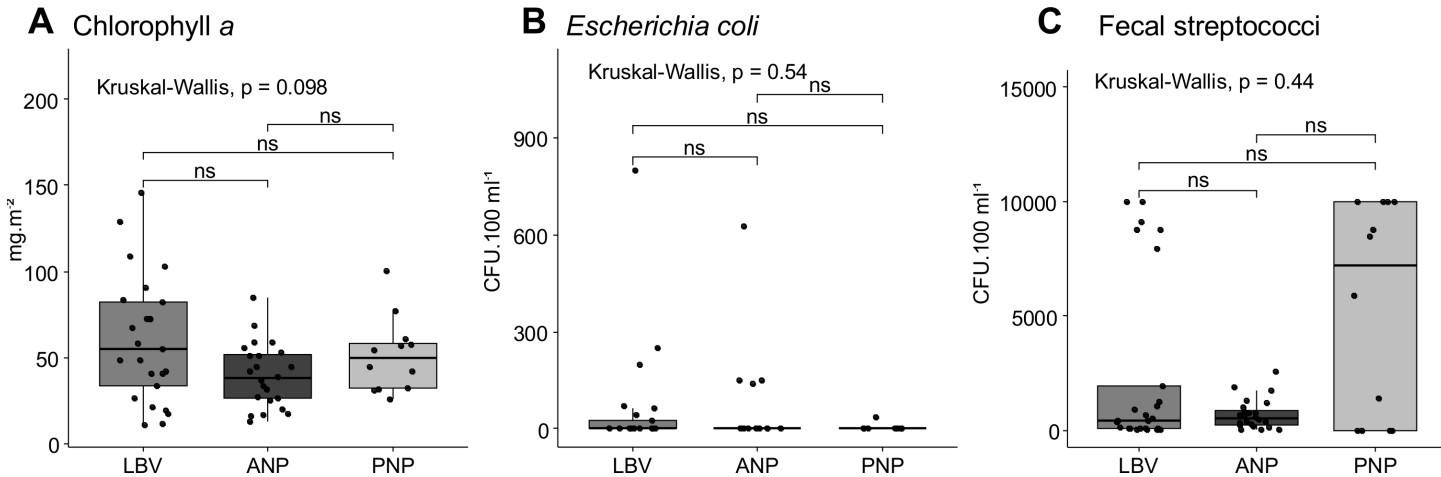

**Fig 4. Biological markers in surface sediments.** Chlorophyll *a* (Chl *a*) surface density (A) expressed in mg.m⁻², and Fecal Indicator Bacteria *Escherichia coli* (E. coli, B) and fecal streptococci (Strept, C) in CFU.100 ml⁻¹ are reported. Kruskal-Wallis p-value is given for the three groups differences; inter-group significant differences (Wilcoxon test) are marked with asterisks (*: $p < 0.05$; **: $p < 0.01$; ***: $p < 0.001$). ns = non-significant difference.

wastewater into the Lowe River ($100.88 \pm 22.13$ mg.m⁻²). In addition, the results indicated a local input of nutrients from sewage discharge and a decreasing gradient from the urban center of Libreville to the ANP along the Ntsini channel, with an exception at Moka Channel stations where Chl *a* surface densities were higher (beach site MKP, $68.19 \pm 16.78$ mg.m⁻²). In the PNP, the mean chlorophyll *a* density pattern exhibited low values with an increasing gradient from the protected sites in the Gongoue River (Fig 1), (upper mangrove river site PG1: $30.74 \pm 2.42$ mg.m⁻²) to the exposed sites in the estuary (Northernmost station PG7: 100.13 mg.m⁻²). However, the paired comparison of the Chl *a* surface densities of the sites didn't show significant differences between Libreville and the two National Parks ($p = 0.051$ and $p = 0.6$ respectively).

*Escherichia coli (E. coli)* was detected across the three sites (Table 2, S4 Table in S1 File), with five outliers in Libreville (LBV), four in ANP and one in PNP with higher CFU counts (Fig 4B). Kruskal-Wallis scores showed no statistically significant difference ($p = 0.59$) in *E. coli* contamination between ANPs, LBVs and PNPs suggesting a possible transfer of sewage from Libreville to protected areas and/ or local source within these latter.

Fecal streptococci were detected at all stations except one in PNP. Contamination levels were not statistically different between sites ($p = 0.43$). As with Chl *a* surface density, Streptococci contamination decreased from urban areas (maximum > 10,000 CFU.100 ml⁻¹) to the Akanda National Park (25–2,563 CFU.100 ml⁻¹) along the Ntsini Channel to downstream in Mondah Bay (S4 Table in S1 File). In PNP, contamination increased from upstream to exposed estuarine stations suggesting an advection of contaminants from Libreville through the Komo estuary due to hydrodynamics.

### Trace metal contamination assessment

**Trace metal concentrations and ecotoxicity potential.** The concentrations of TMs in sediment and their distribution across the sites are illustrated in Fig 5. Libreville (LBV) exhibits the highest variability in As (0.26 to 14.09 µg.g⁻¹), Cd (0.01 to 0.26 µg.g⁻¹), and Cr (1.75 to 90.13 µg.g⁻¹) concentrations, with three outliers for Cu, two for Pb, and one for Zn. Among the parks, Akanda National Park (ANP) displays the widest concentration ranges for As (7.03 to 13.40 µg.g⁻¹) and Cr (68.0 to 193.4 µg.g⁻¹), while Cu (7.95 to 160.83 µg.g⁻¹), Pb (13.90 to 168.34 µg.g⁻¹), and Zn (44.28 to 1,083.94 µg.g⁻¹) generally remain at lower levels, except for two outliers in Pb and Zn and one in Cu. By contrast, Pongara National Park has the highest concentration range of Cd (0.13 to 0.23 µg.g⁻¹), with single outliers observed in Cu (350.80 µg.g⁻¹), Pb (714.22 µg.g⁻¹), and Zn (3,569.29 µg.g⁻¹). All the data, expressed in µg/g dry weight, are provided as supplementary

**Table 2. Observed ranges of biological markers. Chlorophyll *a* (mg.m⁻²) and Fecal Indicator Bacteria *Escherichia coli* (*E. coli*, B) and fecal streptococci (Strept, C) in CFU.100 ml⁻¹.**

| Sites | Chl *a* (mg.m⁻²) | *E. coli* (CFU.100 ml⁻¹) | Strept (CFU.100 ml⁻¹) |
|---|---|---|---|
| LBV | 10.94−145.93 | 0−800 | 0−10,000 |
| ANP | 13.20−84.8 1 | 0−625 | 25−2,563 |
| PNP | 25.57−100.12 | 0−35 | 0−10,000 |

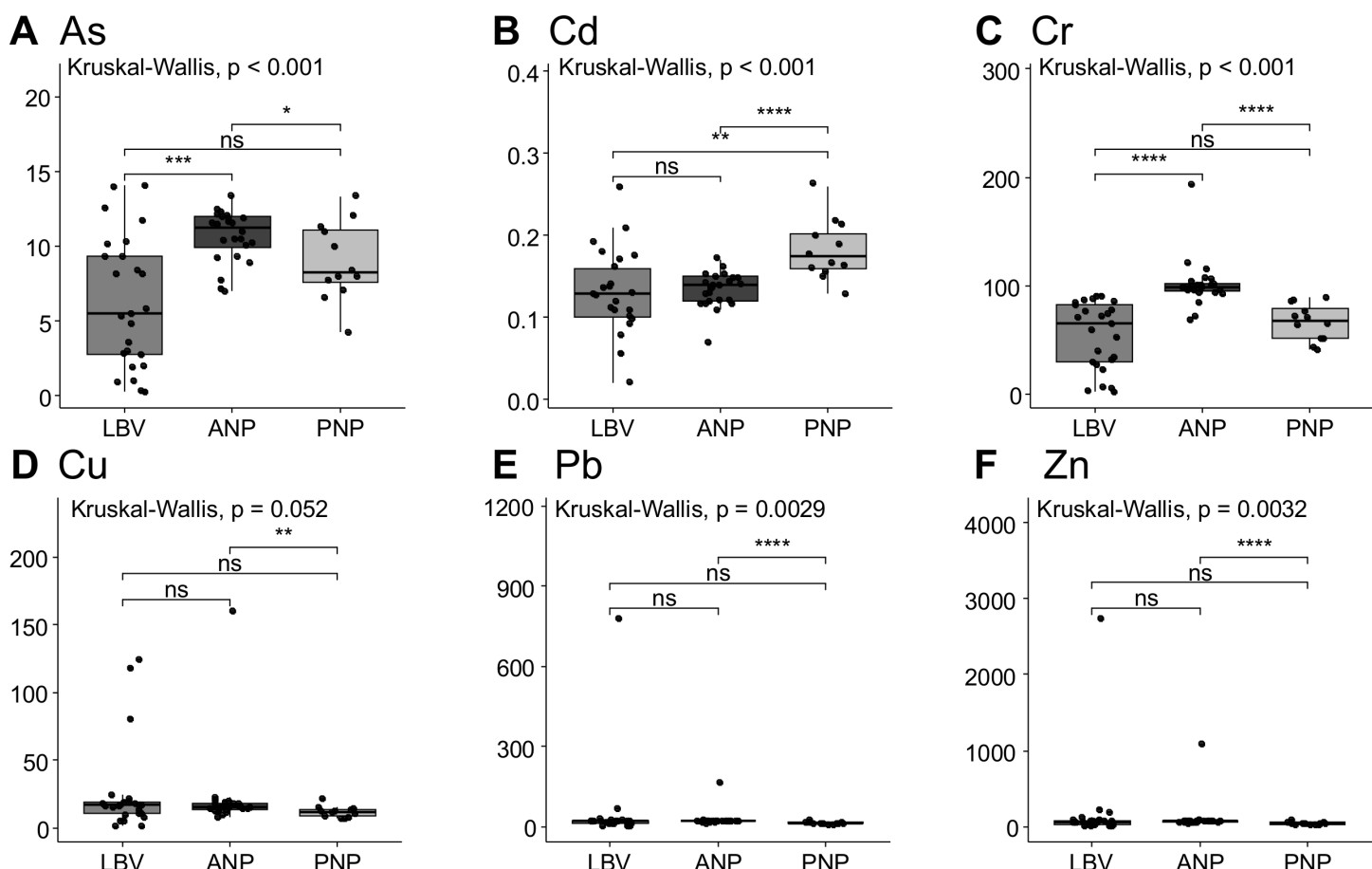

**Fig 5. Trace metal concentrations in surface sediments.** Variability of trace metal concentrations (in µg.g⁻¹) in surface sediments in LBV, ANP and PNP. Arsenic (A), cadmium (B), chromium (C), copper (D), lead (E) and zinc (Zn) are expressed in µg.g⁻¹ dry sediment. Kruskal-Wallis p-value is given for the three groups differences; inter-group significant differences (Wilcoxon test) are marked with asterisks (*: p < 0.05; **: p < 0.01; ***: p < 0.001). ns = non-significant difference.

material (S5 Table in S1 File), together with concentrations of TMs cobalt, manganese, nickel, tin, the alkali metal lithium, and the actinide uranium (S6 Table in S1 File).

The concentration of Cu differs significantly particularly between ANP and PNP (p = 0.028). Our results indicate significant differences in As and Cr concentrations between Libreville and ANP (p < 0.001), in Cd, Cr, Cu, Pb and Zn between the two parks (p < 0.001 except for Cu, p = 0.005), but no statistical difference between Libreville and PNP (p > 0.08) in As,

Cr, Cu, Pb and Zn (for Cd, p = 0.002). However, the presence of outliers, particularly for Cu, Pb, and Zn, suggests high variability and site-specific contamination conditions in both Libreville and the parks.

The potential biological impacts of TMs on marine organisms were assessed using two sets of sediment quality guidelines (SQGs): TEL/PEL and ERL/ERM. Cd concentrations at all three sites were below the four toxicity thresholds (Table 3). The maximum As concentrations remained below the PEL and ERM toxicity levels across all sites. At ANP, Cr concentrations exceeded the TEL, ERL, and PEL thresholds but remained below the ERM, whereas at LBV and PNP, they were higher than the TEL and ERL but below both the PEL and ERM guidelines. Maximum Cu concentrations exceeded toxicity thresholds, suggesting frequent negative impacts at ANP and Libreville, except in Pongara National Park, where the highest concentrations surpassed the ERM threshold. For Pb and Zn, the highest recorded levels at each site exceeded all sediment quality guideline toxicity thresholds, except at ANP, where the highest Pb concentration remained below the ERM toxicity value.

**Assessment of trace metal contamination sources.** The contamination levels of mudflat sediments were assessed using two indices calculated for the six target TMs. The EF and $I_{geo}$ followed the order: Zn > Pb > Cu > Cr > As > Cd. The EF results (Fig 6A) indicate no enrichment (EF < 1) to low enrichment (1 < EF < 3) for As, Cd, Cr, and Cu across the three sites. However, the EF analysis reveals severe Cu enrichment (EF = 23.12) at a station in Libreville, significantly influencing the overall average EF value. Consequently, the urban area of Libreville exhibits high to extreme Pb and Zn enrichment, whereas sediments in the parks show only low Pb and Zn enrichment (EF < 3).

The $I_{geo}$ results (Fig 6B) indicate that sediments from all three sites are unpolluted by As, Cd, Cr, and Cu (class 0, $I_{geo}$ < 0), while 83.3% of the samples are unpolluted by Pb and Zn. However, sediments from Libreville and ANP range from unpolluted to moderately polluted (class 1, 0 < $I_{geo}$ < 1). In contrast, stations in the Libreville urban area and PNP exhibit moderate to high pollution levels (2 < $I_{geo}$ < 3). Most sites show moderate to severe Zn pollution (1 < $I_{geo}$ < 5), except at PNP, where $I_{geo}$ indicates no to moderate pollution (0 < $I_{geo}$ < 1) in most samples.

## Sediment characterization using multiple factors

In order to improve the characterization of contaminated sediments and to identify the main sources of variability, we applied multivariate approaches incorporating different parameters. In the PCA results (Fig 7), the first two principal components account for about 58% of the total variance in sediment characterization (PC1: 33.3%, PC2: 24.7%).

Table 3. Trace metals range values compare to sediment quality guidelines. Sediment Quality Guidelines (SQGs) are reported for each of the six trace metals: threshold effect level (TEL), potential effect level (PEL), effect range-low (ERL), and effect range-medium (ERM). Ranges (min-max for the whole survey) of measured TMs concentrations in surface sediments from ANP, LBV, and PNP are compared, values exceeding PEL are highlighted in bold.

| SQGs | TMs concentrations (µg.g⁻¹ dry weight) | | | | | |
|---|---|---|---|---|---|---|
| | As | Cd | Cr | Cu | Pb | Zn |
| TEL | 7.2 | 0.6 | 52 | 18.7 | 30.2 | 124 |
| PEL | 41.6 | 4.2 | 160 | 108.0 | 112.0 | 271 |
| ERL | 8.2 | 1.2 | 81 | 34.0 | 47.0 | 150 |
| ERM | 70.0 | 9.6 | 370 | 270.0 | 218.0 | 410 |
| | Range values in the present study | | | | | |
| LBV | 0.3–14.1 | 0.0–0.3 | 1.7–90.1 | 1.3–**124.4** | 2.5–**780.8** | 4.8–**2745.7** |
| ANP | 7.0–13.4 | 0.0–0.2 | 69.0–**193.4** | 7.9–**160.8** | 13.9–**168.3** | 44.3–**1083.9** |
| PNP | 4.3–13.5 | 0.1–0.3 | 40.4–88.7 | 6.9–**350.8** | 6.7–**714.2** | 26.1–**3569.3** |

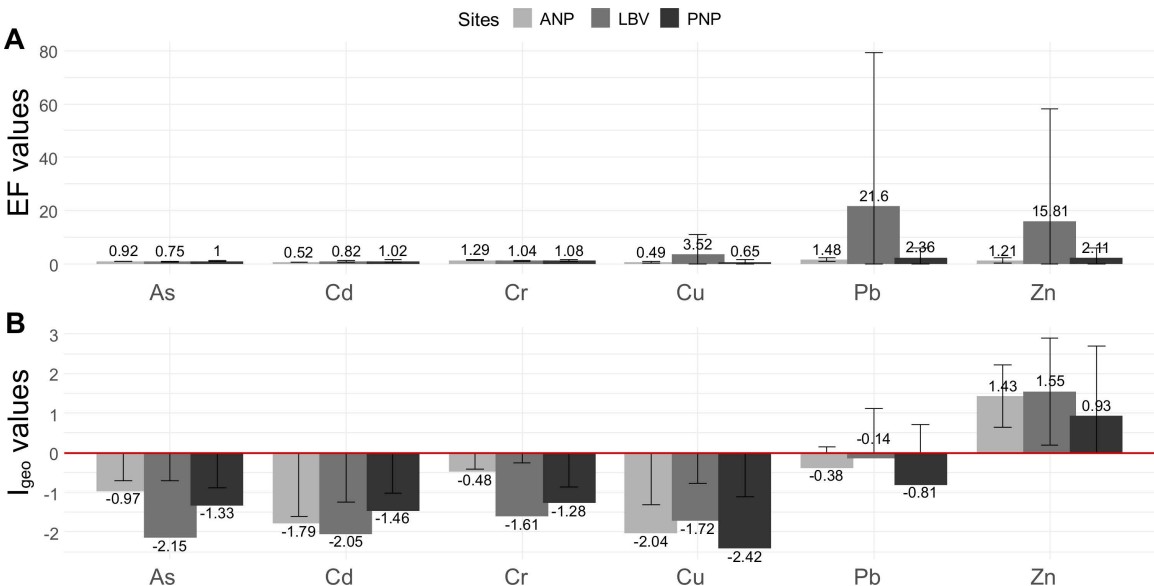

**Fig 6. Trace metal contamination indices.** Enrichment Factor (A) and (B) Geo-accumulation index I_geo (B) are provided for the six target TMs for the three sites LBV, ANP and PNP.

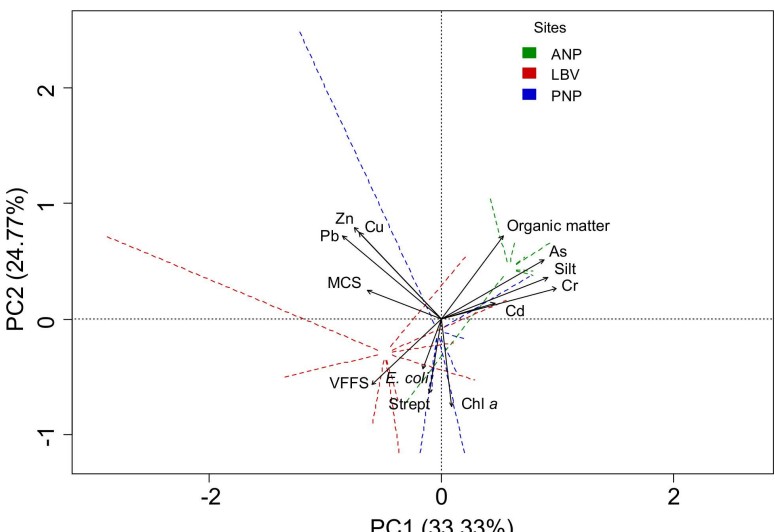

**Fig 7. Multivariate analysis of trace metal in sediments.** Mangrove mudflat sediments from ANP, LBV, and PNP were characterized by Principal Component Analysis, based on biological markers (FIB and Chl *a*) and trace metal concentrations.

PC1, which explains most of the variability, distinguishes two groups: on the one hand, sandy stations from LBV and PNP, highly polluted by Zn, Pb and Cu likely due to anthropogenic activities; on the other hand, ANP, characterized by high organic matter and silt, with As, Cd and Cr originating from crustal sources.

The second principal component, PC2, contrasts TMs, organic matter, and silt with very fine sand, FIB, and Chl *a*. However, Pearson correlation analysis (Fig 8) reveals a positive but non-significant relationship between Chl *a*, FIB, and very fine sand content of sediments. This may suggest a common contamination source for FIB and nutrients favoring Chl *a*

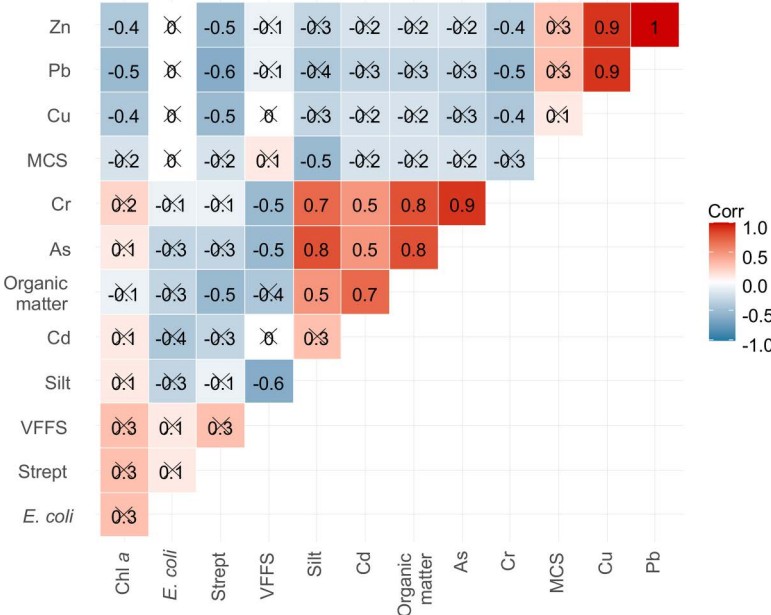

**Fig 8. Relationship between sediment characteristics, biological markers and trace metal concentrations.** Pearson correlation coefficients of biological and chemical contamination markers and soil properties are reported; threshold values are 0.4 and −0.4 for positive (red) and negative (blue) correlations, respectively.

and biofilm development. Our findings confirm the affinity between trace metals and mud, as well as the potential cooccurrence of fine sand with high surface chlorophyll density and FIB concentration.

Additionally, the nonparametric multivariate analysis of variance (npmv), based on ANOVA-type test statistics, reveals a significant distinction between the three sites in terms of contamination levels by multiple contaminant markers ($p < 0.001$). The npmv results reject the hypothesis of equal contamination levels between ANP and PNP, as well as between ANP and LBV ($p < 0.001$). However, the hypothesis is not rejected for LBV vs. PNP (error rate at alpha = 0.05), suggesting a similar contamination pattern between the urban area and the protected site.

## Discussion

### Sediment properties

Sediment profiles varied significantly across the three sites. These differences could be attributed, on one hand, to stronger hydrodynamic conditions in the Komo Estuary compared to the Mondah, where slower water flow favors the deposition of finer particles such as silt [53]. On the other hand, they may result from multiple sources of organic matter enrichment, including proximity to mangrove trees. As reported by [54], organic matter in mangrove sediments originates from various sources, including mangrove litter and imported suspended material from terrestrial, micro- and macroalgal, riverine, and marine inputs. Consequently, the composition of organic matter in mangrove sediments depends on surrounding tree species [55] and weathering conditions [56].

### Biological pollution markers

Surface Chlorophyll *a* (Chl *a*) density, an indicator of eutrophication, decreases from Libreville to ANP, suggesting urban river discharge influence. High Chl *a* levels are typical of eutrophic, human-impacted environments [57,58]. Our

results showed lower Chl *a* density in muddy sediments, except in one deforested station in Libreville (97% sand, Chl *a* 19.75 ± 6.15 mg.m$^{-2}$). Hydrodynamics and microphytobenthic communities may also influence local Chl *a* density [59].

While no health standards exist for sedimentary FIB contamination [20], *E. coli* levels in PNP sediments were below the recreational water threshold of 100 CFU.100 mL$^{-1}$. At ANP, one station exceeded this level (208 ± 360 CFU.100 mL$^{-1}$), while in Libreville, two urban stations showed higher contamination (275 ± 454 and 150 ± 132 CFU.100 mL$^{-1}$). High *E. coli* levels in ANP likely result from a fishermen settlement near Moka Beach, whereas urban contamination is linked to unregulated sewage discharge near docks. *E. coli* tends to be higher in muddy, low-energy environments with high organic matter [20], as seen in ANP.

Fecal streptococci contamination in PNP sediments may stem from both sewage advection and wildlife, including almost all vertebrates. Like Chl *a*, streptococci concentrations were higher at exposed estuarine sites, highlighting the Komo estuary's role in pollutant dispersion. Contamination patterns appear site-specific rather than influenced by MPA status. Streptococci counts were up to 36 times higher than *E. coli* (max 10,000 CFU.100 mL$^{-1}$), consistent with previous studies in similar ecosystems [14,20]. Their greater persistence in coastal environments suggests their higher tolerance to brackish water environment compare to *E. coli*, making them a more reliable indicator of fecal pollution in estuaries [20,60].

### Trace metals concentrations, contamination assessment and sources

Trace metal concentrations in mudflat sediments followed the order Zn > Pb > Cu > Cr > As > Cd. These findings are in agreement with the contamination levels of macroorganisms in the study area [15], where Zn and Cu were the most concentrated metals, and Cd was the least in marine organisms from Akanda Park (ANP). The copper concentrations at ANP (21.44 ± 16.90 µg.g$^{-1}$), LBV (23.79 ± 16.54 µg.g$^{-1}$), and PNP (35.53 ± 59.92 µg.g$^{-1}$) were higher than in comparable environments (Table 4). Zinc concentrations at all three sites were higher than those reported for example in Senegal mangroves [61], Ghana wetlands [17], Brazilian [62] or Malaysian [63] mangroves. PNP sediments showed the highest Pb concentrations (63.76 ± 122.17 µg.g$^{-1}$) and Zn (297.24 ± 617.26 µg.g$^{-1}$), though Peninsular Malaysia had higher Pb levels (47.87 ± 47.15 µg.g$^{-1}$) than ANP (28.00 ± 16.64 µg.g$^{-1}$). Almeida et al. [64] have also identified Zn and Cu as the most hazardous pollutants in mangrove sediments from the Joanes River, Brazil. Chromium concentrations in ANP (101.59 ± 6.55 µg.g$^{-1}$) were higher than in other sites, while PNP (63.31 ± 12.97 µg.g$^{-1}$) had lower levels than those in the Songor Wetland in Ghana (66.00 ± 15.00 µg.g$^{-1}$) and estuaries near Recife, Brazil (61.2 ± 14.23 µg.g$^{-1}$). In contrast, Recife (15.86 ± 7.98

**Table 4. Mean concentrations of trace metals in surface mudflat sediments from ANP, LBV, and PNP (Gabon) compared with results from other tropical estuaries (µg/g). n.a. = not available.**

| Sites | As | Cd | Cr | Cu | Pb | Zn | References |
|---|---|---|---|---|---|---|---|
| ANP | 10.69 ± 1.56 | 0.13 ± 0.01 | 101.59 ± 6.55 | 21.44 ± 16.90 | 28.00 ± 16.64 | 114.83 ± 112.46 | Present study |
| LBV | 6.41 ± 3.74 | 0.13 ± 0.05 | 57.26 ± 27.13 | 23.79 ± 16.54 | 41.91 ± 55.55 | 142.09 ± 196.50 | Present study |
| PNP | 8.66 ± 2.16 | 0.18 ± 0.03 | 63.31 ± 12.97 | 35.53 ± 59.92 | 63.76 ± 122.17 | 297.24 ± 617.26 | Present study |
| Bundu Creek, Nigeria | 0.014 | 2.92 | n.a. | 533.65 | 8.91 | 223.62 | [66] |
| Songor Wetland, Ghana | 4.25 ± 1.18 | 0.023 ± 0.01 | 66.00 ± 15.00 | 21.80 ± 5.25 | 11.17 ± 2.42 | 40.00 ± 8.00 | [17] |
| Ebrié Lagoon, Ivory Coast | n.a. | 3.77 ± 0.79 | n.a. | 5.74 ± 0.33 | 96.55 ± 9.88 | 157.85 ± 152.15 | [65] |
| Saloum Delta, Senegal | n.a. | 0.03 ± 0.02 | 25.06 ± 4.68 | 2.84 ± 0.76 | 2.31 ± 0.56 | 4.34 ± 1.41 | [61] |
| Recife estuaries, Brazil | 15.86 ± 7.98 | 0.06 ± 0.02 | 61.2 ± 14.23 | 12.14 ± 3.88 | 15.08 ± 2.51 | 45.00 ± 8.79 | [62] |
| Peninsular Malaysia | 34.82 ± 11.55 | 1.57 ± 0.35 | 43.29 ± 13.70 | 16.36 ± 8.23 | 47.87 ± 47.15 | 78.41 ± 35.82 | [63] |

Outliers in Cu, Pb, and Zn at several specific stations, KEN in ANP (Cu = 160.83 µg.g$^{-1}$, Pb = 168.34 µg.g$^{-1}$, Zn = 1,083.93 µg.g$^{-1}$), PG1 in PNP (Cu = 350.80 µg.g$^{-1}$, Pb = 714.21 µg.g$^{-1}$, Zn = 3,569.29 µg.g$^{-1}$), and LW3 in LBV (Pb = 780.83 µg.g$^{-1}$, Zn = 2,745.69 µg.g$^{-1}$) significantly affect the average concentrations across the sites.

μg.g$^{-1}$) and Peninsular Malaysia (34.82 ± 11.55 μg.g$^{-1}$) mangrove sediments exhibited higher As levels. Sediments from the Vridi Canal (Ebrié Lagoon, Ivory Coast) showed the highest Cd enrichment (3.77 ± 0.79 μg.g$^{-1}$) [65]. Cadmium was also reported to be the most prevalent heavy metal in mangrove sediments from the Bonny Estuary (Bundu Creek), Nigeria [66].

The Geo-accumulation index ($I_{geo}$) identifies contamination sources, with values < 0 indicating natural sources and > 1 indicating anthropogenic contamination, while the Enrichment Factor (EF) evaluates the impact of anthropogenic activities on metal concentrations. Negative $I_{geo}$ values and low EF (EF < 3) suggest natural contamination from the crust. However, the EF reveals severe Cu enrichment (EF = 23.12) at Libreville's LW3 station, strongly influencing the average EF value. Libreville's central urban area (LW1 and LW3) shows high Pb and Zn enrichment, especially at LW3, where Pb and Zn are extremely enriched. EF and $I_{geo}$ values for Cu, Pb, and Zn suggest anthropogenic sources, such as industrial effluents, fossil fuel combustion, and traffic [67,68]. The extreme enrichment at LW3 may be due to low organic matter, high sand content, and proximity to petroleum service stations, while the parks enrichment likely results from fishing boat traffic.

## Sediment characterization using multiple factors

Contamination levels assessed using EF and $I_{geo}$ can vary significantly depending on the selected background values [17,43], particularly for Cr, Cu, and Pb [69]. Multivariate techniques, including correlation and PCA, enhance the characterization of contaminated sediments and the identification of key variability sources in FIB [14]. The significant correlations between Zn, Pb, and Cu suggest a shared anthropogenic source, likely linked to fossil fuel discharge and boat traffic in both the city and parks.

In Libreville, stations downstream of the Mindoubé open landfill, near a fishing camp, gas station, and market dock showed higher trace metal concentrations. In PNP, station PG1, located on steep banks near a former fishing village, exhibited high Cu, Pb, and Zn levels. The positive correlation between trace metals, silt, and organic matter is attributed to the fine grain fraction's higher surface area and cation exchange capacity [70,71] and the complexation of metals by organic matter [56,64].

These findings suggest that intra-site variability is more significant than differences between the protected area and the city, with contamination being localized in specific areas under certain conditions. Estuarine hydrodynamics also play a role in pollutant distribution. Untreated wastewater discharges into waterways, reaching the parks and river channels. At low tide, contaminants settle in sediments, and at high tide, they remobilize, posing environmental risks, especially to the food chain and groundwater [72]. Uncontrolled urban expansion of Libreville outskirt further reduces buffer zones around protected areas, while the absence of wastewater management increased the direct exposure to contaminants [73]. Strengthening urban planning regulations and improving wastewater treatment should be priorities for reducing pollutant pressures near MPAs. However, downsizing or downgrading of PAs remains mainly driven by resource exploitation and local pressures rather than regulatory weaknesses [12].

## Conclusion

This study provides baseline data on chlorophyll *a* (Chl *a*), fecal indicator bacteria (FIB), and trace metal concentrations in mangrove mudflat sediments from 24 stations in two marine protected areas and Libreville, Gabon. Biological markers showed higher contamination in human-impacted areas and lower levels in muddy sediments at Akanda National Park. Pongara National Park contamination pattern, similar to Libreville, suggests a possible direct influence of urban leaching to the protected ecosystem. Trace metal concentrations of Cu, Pb, and Zn were higher than in several other West African estuaries, while As, Cd, and Cr levels varied more. Geochemical indices (EF and $I_{geo}$) revealed natural enrichment for As, Cd, and Cr, and anthropogenic enrichment for Cu, Pb, and Zn, likely from heavy traffic and fuel discharges. The large natural variability in conditions made it complex to assess anthropogenic impact, and spatial differences were more related to station characteristics than the actual protection status. These results highlight the importance of assessing the exposure

of marine protected areas to pollution, whether already established or intended. Although sediment contamination levels in parks around Libreville appeared generally low in 2023, the scope of our analysis was not exhaustive. Future studies are recommended to include additional contaminants, particularly persistent organic pollutants (POPs) and microplastics, given that Libreville is home to two rivers (Batavea and Oloumi) both ranked among the 1,000 most plastic-polluted worldwide [74], despite their short watercourse. Moreover, Hg was not assessed in the present study, despite its widespread use in artisanal gold mining across Gabon. Expanding the range of contaminants considered into future monitoring efforts would provide a more comprehensive understanding of environmental pressures in protected areas. This is especially important when these MPAs are delimited close to anthropogenic sources of pollution such as urban centers, and represent a significant biological resource through artisanal fisheries. By highlighting a range of localized pressures, our results underscore the need to adopt a multi-contaminant approach in environmental assessment. This would help bridge current scientific knowledge gaps and improve the effective management of MPAs.

## Supporting information

**S1 File. S1 Table**. Sampling locations. Three-digit codes were designed according to local names (in French). Environment type and ESPG 4326 – WGS 84 coordinates are provided for each sampling location. **S2 Table**. Sampling schedule during the survey. ANP: Akanda; LBV: Libreville; PNP: Pongara National Park; sampling location codes are reported in S1 Table and figured in Fig 1. of the main manuscript. **S3 Table**. Sediment properties. Sediment in situ temperature and physico-chemical properties. **S4 Table.** Markers of biological pollution. Mean Fecal indicator bacteria (FIB) counts are given as colony-forming units (CFU), and mean chlorophyll *a* density in mg.m$^{-2}$ for each sampling site during the whole survey. **S5 Table.** Trace metal concentrations in the sediments. The six trace metals retained for pollution assessment are Arsenic (As), Cadmium (Cd), Chromium (Cr), Copper (Cu), Lead (Pb) and Zinc (Zn), concentrations are given in µg.g$^{-1}$ (dry weight of sediment). Iron (Fe) content in in µg.g$^{-1}$ (dry weight of sediment) is reported as reference metal for Enrichment Factor determination. **S6 Table.** Concentration of elements quantified by ICP-MS, not included in assessment. Concentrations are given in µg.g-1 (dry weight of sediment).
(ZIP)

## Acknowledgments

This research was conducted under CNAR agreements AR022/21 and AR043/22, granted by CENAREST and approved by the Agence Nationale des Parcs Nationaux (ANPN) (work permits AE22008 and AE220017), Republic of Gabon. The authors thank the Gabonese authorities for their administrative support, particularly National Park curators M. Gilbert Moukanga (Akanda NP), M. Luc Patrick Evezo'o (Pongara NP), and ANPN staff for their help with field sampling. Special thanks to Romain Gonzalvez for his help preparing samples for ICP-MS analysis, and to Professor Makaya M'voubou and Dr. Mokea-Niaty for their hospitality and assistance during granulometric and organic matter content analyses at the USTM polytechnic school and Biology Department laboratories.

## Author contributions

**Conceptualization:** Aimé Roger Nzigou, Gauthier Schaal, Christophe Leboulanger, Patrick Mickala.

**Data curation:** Marie-Laure Rouget.

**Formal analysis:** Johann Ludovic Martial Happi, Aimé Roger Nzigou, Marie-Laure Rouget, Christophe Leboulanger.

**Funding acquisition:** Christophe Leboulanger.

**Investigation:** Johann Ludovic Martial Happi, Aimé Roger Nzigou, Christophe Leboulanger.

**Methodology:** Johann Ludovic Martial Happi, Christophe Leboulanger.

**Resources:** Jean-Daniel Mbega.

**Supervision:** Christophe Leboulanger, Patrick Mickala.

**Validation:** Aimé Roger Nzigou, Gauthier Schaal, François Le Loc'h.

**Writing – original draft:** Johann Ludovic Martial Happi.

**Writing – review & editing:** Johann Ludovic Martial Happi, Aimé Roger Nzigou, Gauthier Schaal, Marie-Laure Rouget, Rolf Gael Mabicka Obame, François Le Loc'h, Jean-Daniel Mbega, Christophe Leboulanger, Patrick Mickala.

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
