## [Decision Letter · Decision Letter 0]

PONE-D-25-11485Characterization of mangrove mudflat sediment contamination by fecal bacteria and trace metals: A multivariate assessment in equatorial marine protected areas in GabonPLOS ONE

Dear Dr. Leboulanger,

Thank you for submitting your manuscript to PLOS ONE. After careful consideration, we feel that it has merit but does not fully meet PLOS ONE’s publication criteria as it currently stands. Therefore, we invite you to submit a revised version of the manuscript that addresses the points raised during the review process.

We look forward to receiving your revised manuscript.

Kind regards,

Linton Munyai, PhD

Academic Editor

PLOS ONE

Journal Requirements:

“PhD stipend to Johann L.M. Happi by the ARTS program of IRD (French National Research Institute for Sustainable Development)”

Please state what role the funders took in the study.  If the funders had no role, please state: 'The funders had no role in study design, data collection and analysis, decision to publish, or preparation of the manuscript.'

“This research is part of a PhD thesis (JLMH) funded by the IRD ARTS scholarship program and conducted under CNAR agreements AR022/21 and AR043/22, granted by CENAREST and approved by the Agence Nationale des Parcs Nationaux (ANPN) (work permits AE22008 and AE220017), Republic of Gabon. The authors thank the Gabonese authorities for their administrative support, particularly National Park curators M. Gilbert Moukanga (Akanda NP), M. Luc Patrick Evezo’o (Pongara NP), and ANPN staff for their help with field sampling. Special thanks to Romain Gonzalvez for his help preparing samples for ICP-MS analysis, and to Professor Makaya M’voubou and Dr. Mokea-Niaty for their hospitality and assistance during granulometric and organic matter content analyses at the USTM polytechnic school and Biology Department laboratories.”

“PhD stipend to Johann L.M. Happi by the ARTS program of IRD (French National Research Institute for Sustainable Development)”

6. We note that Figure 1 in your submission contain [map/satellite] images which may be copyrighted. All PLOS content is published under the Creative Commons Attribution License (CC BY 4.0), which means that the manuscript, images, and Supporting Information files will be freely available online, and any third party is permitted to access, download, copy, distribute, and use these materials in any way, even commercially, with proper attribution. For these reasons, we cannot publish previously copyrighted maps or satellite images created using proprietary data, such as Google software (Google Maps, Street View, and Earth). For more information, see our copyright guidelines: http://journals.plos.org/plosone/s/licenses-and-copyright.

Please upload the completed Content Permission Form or other proof of granted permissions as an ''Other'' file with your submission.

Reviewers' comments:

Reviewer's Responses to Questions

**Comments to the Author**

1. Is the manuscript technically sound, and do the data support the conclusions?

Reviewer #1: Partly

Reviewer #2: Yes

2. Has the statistical analysis been performed appropriately and rigorously? 

Reviewer #1: Yes

Reviewer #2: Yes

3. Have the authors made all data underlying the findings in their manuscript fully available?

Reviewer #1: Yes

Reviewer #2: Yes

4. Is the manuscript presented in an intelligible fashion and written in standard English?

Reviewer #1: No

Reviewer #2: Yes

5. Review Comments to the Author

Reviewer #1: The manuscript presents a potentially valuable study on pollution levels in mangroves within Gabon. However, to enhance its impact and accessibility, I recommend revisions addressing the contextual information provided, the justification for certain conclusions, and the overall flow of the narrative.

Specific Comments:

• I strongly suggest modifying the title to include geographical identifiers beyond just "Gabon." Specifying "Gabon, Africa" or "Gabon, Central Africa" will immediately orient readers unfamiliar with the country's location. This small change will significantly broaden the audience who can quickly grasp the study's focus.

• The abstract should begin by establishing the broader significance of mangroves, their global distribution, and their critical role in coastal ecosystems. Quantify their area cover globally and within Gabon if possible. Furthermore, clearly articulate the vulnerability of mangroves to pollution, emphasizing the potential consequences for biodiversity and ecosystem services. Then, explicitly connect this overarching context to the specific research question and approach undertaken in the study. This will highlight the relevance and importance of your work.

• Introduce the primary anthropogenic threats faced by Gabon's mangrove ecosystems in the abstract. Mention specific activities like oil and gas extraction, urbanization, or deforestation, as applicable. Briefly explain how these threats might contribute to the observed pollution levels and how your findings inform potential mitigation strategies. This helps readers understand the real-world implications of your research.

• Expand upon the role of Marine Protected Areas (MPAs), specifically the MNP mentioned in the manuscript. Explain the rationale behind designating these areas as MPAs. Detail the presence of Endangered, Threatened, and Protected (ETP) species within the MNP, highlighting ecologically sensitive zones within the protected area. This provides crucial context for understanding the potential impacts of pollution on vulnerable species and habitats and underlines the need for targeted conservation efforts. Discuss the ecological significance of mangroves harboring diverse marine fauna and how pollutants might negatively impact them.

• Justification for Conclusions on MPA Effectiveness (Lines 36-40): The conclusion that MPAs are not effective based on the observed low contamination levels requires substantial justification. This statement seems premature and potentially misleading without further analysis. I recommend carefully re-evaluating this conclusion and providing a more nuanced interpretation. Consider the possibility that the existing contamination levels are lower than they would be without the MPAs, even if they are not zero. The existing MPA may contribute to preventing further contamination, which this experiment would not test. Acknowledge the limitations of the study design in assessing the effectiveness of the MPAs.

• Include a literature review of case studies where MPAs have demonstrably reduced pressure on mangrove tidal flats. If such studies exist, cite them to provide a broader context for your findings. If studies show limited effectiveness in some contexts, acknowledge these and discuss potential reasons, such as proximity to industrial discharge or inadequate enforcement of regulations. Provide details regarding the proximity of industrial discharge to the studied MNPs and the number of industries located nearby. This information is crucial for evaluating the potential sources of pollution and assessing the effectiveness of the MPAs.

• Similar to the abstract, the introduction should begin by introducing Gabon and emphasizing its significance, particularly in terms of its biodiversity, coastal ecosystems, and role in the regional economy. Provide key geographical and ecological information that is relevant to the study.

• Lines 52-53 mention important coastal wetlands. Provide more detailed information about these wetlands, either in a table or in the text. Include details like their size, ecological characteristics, and any existing conservation status.

• Expand the introduction to include specific information about Gabon's mangroves. Discuss their species diversity, area cover, and ecological importance. Link this information back to lines 52-53, highlighting the importance of these mangroves within the broader context of Gabon's coastal wetlands. Providing statistics and key facts about these mangrove ecosystems will significantly strengthen the introduction.

Methodology:

• Relocation of Lines 102-113: I agree that the information presented in lines 102-113 would be more appropriate within the introduction, as it provides background context relevant to the study site selection and environmental conditions.

• Clarity on Data Collection Campaigns: Rephrase the term "data collection campaigns" to provide a clearer explanation of the specific activities involved. Describe the frequency, duration, and specific parameters measured during these campaigns.

• Table 4 Terminology: Change "This work" to "Present study" in Table 4 for consistency and professionalism.

Discussion and Conclusion:

• Management Recommendations: The discussion and conclusion should include practical recommendations for management measures that can be implemented to protect Gabon's mangroves. These recommendations should be based on the observations and results of the present study. Consider suggesting specific actions related to pollution control, MPA management, or community engagement.

By addressing these points, the authors can enhance the clarity, context, and impact of their research. The revisions will not only make the manuscript more accessible to a broader audience but also strengthen the conclusions and provide valuable insights for mangrove conservation in Gabon. The research has the potential to contribute significantly to the understanding of anthropogenic impacts on coastal ecosystems and inform effective conservation strategies.

Reviewer #2: Nice paper. Important work regarding mangrove pollution.

I have only identified unformatted text in the lines 412 and 413, item "4.3. Trace metals concentrations, contamination assessment and sources".

6. PLOS authors have the option to publish the peer review history of their article (what does this mean? ). If published, this will include your full peer review and any attached files.

**Do you want your identity to be public for this peer review?** For information about this choice, including consent withdrawal, please see our Privacy Policy .

Reviewer #1: No

Reviewer #2: **Yes: ** Fernando de Figueiredo Porto Neto

---

## [Author Response · Author response to Decision Letter 1]

8 May 2025

Dear Editor, Dear Reviewers,

We warmly acknowledge your comments and suggestions. The manuscript has been carefully modified to ensure that all your input contributes to its interest, and that it will finally deserve acceptance for publication in PLOS One.

For convenience, our responses are highlighted in blue, in boxed paragraph, and reference to the revised manuscript line numbers provided accordingly.

Editor’s comments:

Journal Requirements:

Style was checked, namely section headings were modified, and “Fig.” replaced by “Fig” throughout the manuscript.

“PhD stipend to Johann L.M. Happi by the ARTS program of IRD (French National Research Institute for Sustainable Development)”

Please state what role the funders took in the study. If the funders had no role, please state: 'The funders had no role in study design, data collection and analysis, decision to publish, or preparation of the manuscript.'

The financial disclosure was modified accordingly:

“Johann L.M. Happi benefited from a PhD stipend through the ARTS Program of IRD (French National Research Institute for Sustainable Development). The Funder had no role in study design, data collection and analysis, decision to publish, or preparation of the manuscript.“

“This research is part of a PhD thesis (JLMH) funded by the IRD ARTS scholarship program and conducted under CNAR agreements AR022/21 and AR043/22, granted by CENAREST and approved by the Agence Nationale des Parcs Nationaux (ANPN) (work permits AE22008 and AE220017), Republic of Gabon. The authors thank the Gabonese authorities for their administrative support, particularly National Park curators M. Gilbert Moukanga (Akanda NP), M. Luc Patrick Evezo’o (Pongara NP), and ANPN staff for their help with field sampling. Special thanks to Romain Gonzalvez for his help preparing samples for ICP-MS analysis, and to Professor Makaya M’voubou and Dr. Mokea-Niaty for their hospitality and assistance during granulometric and organic matter content analyses at the USTM polytechnic school and Biology Department laboratories.”

“PhD stipend to Johann L.M. Happi by the ARTS program of IRD (French National Research Institute for Sustainable Development)”

Funding-related text was removed from the Acknowledgement section accordingly.

All the data used are currently available without restriction in the Public Repository of the French Research Institute for Sustainable Development (https://dataverse.ird.fr/). Data were curated by the dedicated staff, provided with metadata and relevant information. The Digital Object Identifier address is: https://doi.org/10.23708/GN6XDK

Metadata will be completed with the full reference of the article after acceptance in PLOS One.

Ethics statement was provided in the ‘Methods’ section: the Gabonese National Committee for Research Authorizations (Comité National des Autorisations de Recherche – CNAR) is the relevant authority and serves as Institutional Review Board in Gabon. Accordingly, as some of the sampling was performed in protected areas, the Gabonese National Agency for National Parks (Agence Nationale des Parcs Nationaux – ANPN) provided specific permits.

The relevant section in the revised manuscript was added at the end of Methods section:

Ethics statement

The research authorization and the work permit in protected areas were granted respectively by CENAREST review board (Comité National des Autorisations de Recherche, CNAR) and the Scientific Unit of ANPN (Gabonese National Agency for National Parks). (Lines 257-260)

6. We note that Figure 1 in your submission contain [map/satellite] images which may be copyrighted. All PLOS content is published under the Creative Commons Attribution License (CC BY 4.0), which means that the manuscript, images, and Supporting Information files will be freely available online, and any third party is permitted to access, download, copy, distribute, and use these materials in any way, even commercially, with proper attribution. For these reasons, we cannot publish previously copyrighted maps or satellite images created using proprietary data, such as Google software (Google Maps, Street View, and Earth). For more information, see our copyright guidelines: http://journals.plos.org/plosone/s/licenses-and-copyright.

Please upload the completed Content Permission Form or other proof of granted permissions as an ''Other'' file with your submission.

Response: the map provided as Fig 1 is a personal work of First Author JLHM using Qgis freeware, based on https://www.openstreetmap.org and https://marineregions.org, and completed with publicly available data issued from Gabonese laws. References are therefore added in the Reference list and stated in the Fig 1 legend. (Akanda National Park: République Gabonaise 2007, Décret n° 608/PR/MEFEPEPN; Pongara National Park: République Gabonaise 2007, Décret n° 618/PR/MEFEPEPN; Arboretum Raponda Walker: République Gabonaise 2012, Décret n°0460/PR/MEF; Marine Reserves: République Gabonaise 2017, Décret N° 00161/PR)

Reviewers' comments:

Reviewer's Responses to Questions

Comments to the Author

1. Is the manuscript technically sound, and do the data support the conclusions?

Reviewer #1: Partly

Reviewer #2: Yes

2. Has the statistical analysis been performed appropriately and rigorously?

Reviewer #1: Yes

Reviewer #2: Yes

3. Have the authors made all data underlying the findings in their manuscript fully available?

Reviewer #1: Yes

Reviewer #2: Yes

4. Is the manuscript presented in an intelligible fashion and written in standard English?

Reviewer #1: No

Reviewer #2: Yes

5. Review Comments to the Author

Reviewer #1:

The manuscript presents a potentially valuable study on pollution levels in mangroves within Gabon. However, to enhance its impact and accessibility, I recommend revisions addressing the contextual information provided, the justification for certain conclusions, and the overall flow of the narrative.

Specific Comments:

• I strongly suggest modifying the title to include geographical identifiers beyond just "Gabon." Specifying "Gabon, Africa" or "Gabon, Central Africa" will immediately orient readers unfamiliar with the country's location. This small change will significantly broaden the audience who can quickly grasp the study's focus.

Many thanks for this suggestion enhancing the readership impact. The title was modified as:

“Characterization of mangrove mudflat sediment contamination by fecal bacteria and trace metals: A multivariate assessment in equatorial marine protected areas in Gabon, Western Central Africa” accordingly.

• The abstract should begin by establishing the broader significance of mangroves, their global distribution, and their critical role in coastal ecosystems. Quantify their area cover globally and within Gabon if possible. Furthermore, clearly articulate the vulnerability of mangroves to pollution, emphasizing the potential consequences for biodiversity and ecosystem services. Then, explicitly connect this overarching context to the specific research question and approach undertaken in the study. This will highlight the relevance and importance of your work.

Thank you for the suggestion. A revised version of the Abstract now reads at the beginning:

“Among the most productive ecosystems worldwide, mangroves contribute to global carbon sequestration and play a pivotal role for many species, including supporting the sustainable provision of intertropical fisheries resources. []” (Lines 29-31)

• Introduce the primary anthropogenic threats faced by Gabon's mangrove ecosystems in the abstract. Mention specific activities like oil and gas extraction, urbanization, or deforestation, as applicable. Briefly explain how these threats might contribute to the observed pollution levels and how your findings inform potential mitigation strategies. This helps readers understand the real-world implications of your research.

These concerns are highly relevant. To meet the constraints of the Abstract, local threats are summarized as follows:

“[] Despite the many essential ecosystem services they provide, mangrove ecosystems are facing increasing anthropogenic pressure, primarily because they develop in littoral areas where human activities are rapidly expanding, causing deforestation, urban extension and pollution. Mangroves cover over 1,700 km² of Gabon's coastline, stretching within protected zones or located alongside populated areas, raising concerns about the potential impact of pollution. This study assessed pollution levels in mangrove surface sediments from 24 stations in the capital city of Libreville and the two adjacent Marine Protected Areas (MPAs), Akanda and Pongara, where human impact has not been fully characterized. []” (Lines 31-38)

• Expand upon the role of Marine Protected Areas (MPAs), specifically the MNP mentioned in the manuscript. Explain the rationale behind designating these areas as MPAs. Detail the presence of Endangered, Threatened, and Protected (ETP) species within the MNP, highlighting ecologically sensitive zones within the protected area. This provides crucial context for understanding the potential impacts of pollution on vulnerable species and habitats and underlines the need for targeted conservation efforts. Discuss the ecological significance of mangroves harboring diverse marine fauna and how pollutants might negatively impact them.

Many thanks for the suggestion. Unfortunately, these topics cannot be in-depth detailed in the Abstract section, but are taken into account in the Introduction (Lines 65-73); emphasis on

---

## [Decision Letter · Decision Letter 1]

Characterization of mangrove mudflat sediment contamination by fecal bacteria and trace metals: A multivariate assessment in equatorial marine protected areas in Gabon, Western Central Africa.

PONE-D-25-11485R1

Dear Dr. Leboulanger,

We’re pleased to inform you that your manuscript has been judged scientifically suitable for publication and will be formally accepted for publication once it meets all outstanding technical requirements.

Kind regards,

Linton Munyai, PhD

Academic Editor

PLOS ONE

Additional Editor Comments (optional):

I'm pleased to inform you that your manuscript has been deemed suitable for publication in PLOS ONE. Congratulations! 

Reviewers' comments:

Reviewer's Responses to Questions

**Comments to the Author**

1. If the authors have adequately addressed your comments raised in a previous round of review and you feel that this manuscript is now acceptable for publication, you may indicate that here to bypass the “Comments to the Author” section, enter your conflict of interest statement in the “Confidential to Editor” section, and submit your "Accept" recommendation.

Reviewer #1: All comments have been addressed

2. Is the manuscript technically sound, and do the data support the conclusions?

Reviewer #1: Yes

3. Has the statistical analysis been performed appropriately and rigorously? 

Reviewer #1: Yes

4. Have the authors made all data underlying the findings in their manuscript fully available?

Reviewer #1: Yes

5. Is the manuscript presented in an intelligible fashion and written in standard English?

Reviewer #1: Yes

6. Review Comments to the Author

Reviewer #1: (No Response)

7. PLOS authors have the option to publish the peer review history of their article (what does this mean? ). If published, this will include your full peer review and any attached files.

**Do you want your identity to be public for this peer review?** For information about this choice, including consent withdrawal, please see our Privacy Policy .

Reviewer #1: No

---

## [Editor Report · Acceptance letter]

PONE-D-25-11485R1

PLOS ONE

Dear Dr. Leboulanger,

I'm pleased to inform you that your manuscript has been deemed suitable for publication in PLOS ONE. Congratulations! Your manuscript is now being handed over to our production team.

Kind regards,

on behalf of

Dr. Linton Munyai

Academic Editor

PLOS ONE